# Implementation of a Bio-Inspired Neural Architecture for Autonomous Vehicles on a Multi-FPGA Platform [note 1]

**DOI:** 10.3390/s23104631

**Published:** 2023-05-10

**Authors:** Tarek Elouaret, Sylvain Colomer, Frédéric De Melo, Nicolas Cuperlier, Olivier Romain, Lounis Kessal, Stéphane Zuckerman

**Affiliations:** 1Laboratoire ETIS, CY Cergy-Paris Université, ENSEA, CNRS, 95000 Cergy-Pontoise, France; sylvain.colomer@cyu.fr (S.C.); frederic.demelo@ensea.fr (F.D.M.); nicolas.cuperlier@cyu.fr (N.C.); olivier.romain@cyu.fr (O.R.); kessal@ensea.fr (L.K.); 2VEDECOM Institute, 78000 Versailles, France

**Keywords:** FPGA, bio-inspired algorithms, Wizarde custom platform, Nvidia Jetson TX2, neural networks, N-LOC, hardware acceleration, GTX transceivers

## Abstract

Autonomous vehicles require efficient self-localisation mechanisms and cameras are the most common sensors due to their low cost and rich input. However, the computational intensity of visual localisation varies depending on the environment and requires real-time processing and energy-efficient decision-making. FPGAs provide a solution for prototyping and estimating such energy savings. We propose a distributed solution for implementing a large bio-inspired visual localisation model. The workflow includes (1) an image processing IP that provides pixel information for each visual landmark detected in each captured image, (2) an implementation of N-LOC, a bio-inspired neural architecture, on an FPGA board and (3) a distributed version of N-LOC with evaluation on a single FPGA and a design for use on a multi-FPGA platform. Comparisons with a pure software solution demonstrate that our hardware-based IP implementation yields up to 9× lower latency and 7× higher throughput (frames/second) while maintaining energy efficiency. Our system has a power footprint as low as 2.741 W for the whole system, which is up to 5.5–6× less than what Nvidia Jetson TX2 consumes on average. Our proposed solution offers a promising approach for implementing energy-efficient visual localisation models on FPGA platforms.

## 1. Introduction

Autonomous vehicles have been at the center of attention of robotics and embedded systems fields in the last decade. Indeed, according to a recent technical report by the National Highway Traffic Safety Administration, 94% of road accidents are caused by human errors [1]. Automated Driving solutions (ADSs) are considered to be reliable solutions and are being implemented with the promise of reducing the accident rates, with an ease-of-use of driving vehicles [2].

Current solutions tend to combine multi-modal sensors, e.g., light detection and ranging (LiDAR) cameras, radars, etc., to perform an end-to-end environmental localisation task, coupled with convolutional neural networks (CNNs) as the core process to take decisions [3]. For example, Gao et al. incorporate CNNs and vision algorithms with LIDARs [4]. However, leveraging LiDAR cameras for localisation is expensive and energy consuming [5].

Hence, alternative bio-inspired architectures have been proposed to resolve the navigation task for self-driving vehicles. In particular, the architecture proposed by Espada et al. [6] solves the localisation part by adapting a model based on the hippocampal system in the mammal’s brain to the vehicle localisation problem. However, a large number of neurons must be instantiated to fully implement the model. Yet, even though the resulting model is large, it uses low complexity operations (gradient, Gaussian, etc.). Therefore, this model is amenable to efficient implementation in hardware using Field-Programmable Gate Arrays (FPGAs). However, while the number of neurons used in the model is quite large, it requires fewer layers compared to a CNN. This reduces both computation and operation complexity. Espada’s initial implementation was carried out purely in software and was running on a high-end general-purpose processor. This shows the potential of such an approach in unknown environments, but as is, this software implementation cannot scale to perform the localisation task in real-time at reasonably high speeds.

FPGA-based implementations of CNNs show they may yield better results than other heterogeneous systems such as CPU + GPU + software combinations, in terms of both performance and energy consumption [7]. With respect to vehicle localisation, the image acquisition and processing part of the algorithm has already been performed using FPGAs by Fiack et al. The results show a drastic increase in performance once specialised hardware is used [8].

In this context, this paper introduces a new computational co-design approach: “N-LOC”, to solve the scalability problem of the software implementation of the bio-inspired neural network for autonomous vehicle. This approach allows one to accelerate the processing with a significant speedup compared to traditional methods and a low footprint implementation. Experiments first conducted on a ZC706 board demonstrated the potential of the proposed solution and perspectives on a scalable FPGA platform are presented.

In this work, we describe a methodology to implement in hardware a bio-inspired algorithm as an FPGA Intellectual Property (IP), in order to speedup the execution of the LPMP model and make it usable in a real-time context for autonomous vehicles. This paper significantly extends the work we described elsewhere [9] and proposes the following contributions:A design to port the LPMP model to the hardware using FPGAs.N-LOC, a hardware implementation of the LPMP bio-inspired model for navigation, including a set of benchmarks to compare the performance of N-LOC with the reference software implementation of LPMP.In particular and compared to our previous published work, we perform a comparison with the original (optimised) implementation running on a low-power, high-end embedded platform (Nvidia’s Jetson TX2).A design for a distributed hardware implementation in order to deal with larger neural networks.

The remainder of this paper is as follows: Section 2 discusses the bio-inspired algorithms that exist in the state of the art and the one we chose for our hardware implementation. In Section 3, we outline the computational supports needed for our hardware implementation. Section 4 presents our bio-inspired neural architecture, N-LOC, a new custom accelerator for the localisation task of an autonomous vehicle. Section 5 presents the Wizarde heterogeneous platform for the implementation of the model. Section 6 extends the duplication of our main application on one FPGA tile, then demonstrates its capability over Wizarde custom platform. Section 7 describes the related work. Finally, we conclude in Section 8.

## 2. Background

To achieve the task of navigation, a self-driving vehicle must successfully handle a large number of problems simultaneously [10]. For example, the system must locate its position, identify practicable pathways, determine potential routes, prevent sources of accidents, etc.

To deal with such a variety of issues, proposed car architectures are generally composed of a wide variety of modules, specialised in solving a reduced number of problems [11]. The information extracted by these modules is successively merged to go from the raw sensor to the action on the vehicle. The pipeline is often very classical and follows a logical order: first the information from the sensors is processed (perception system); then this information is used to localise the vehicle in its environment (localisation system); thereafter a trajectory is computed from the location of the vehicle (path planning system); finally, the trajectory is read and carried out via motor control mechanisms (motor control system) [12].

In such an architecture, the responses obtained via the localisation modules have a great impact on the performance of the system. Indeed, the planning block relies heavily on the location provided by the system and requires a high degree of reliability [13]. To reach the highest possible level of performance, location blocks are generally based on the use of very powerful and accurate sensors such as LiDAR or RTK GPS [14]. The vehicles that have achieved the greatest navigation trajectories are mostly based on these technologies [15]. For example, the autonomous car proposed by the VisLab team in the VIAC project (VisLab Intercontinental Autonomous Challenge (the objective of the VIAC project was to test the capability of an autonomous vehicle at very high intensity by taking it on a track of 16,000 km from Italy to China)) used information from GNSS and LiDAR to locate the vehicle [16].

However, these sensors remain costly, energy consumptive and heavily impacted by the environment. For example, GNSS is heavily impacted by the nature of the environment and may not function properly around large buildings or in overcast areas. In the context of *electric* (i.e., battery-powered) autonomous vehicles, this yields a significant impact. This has led to the development of new alternatives, such as Visual Place Recognition (VPR) methods [17]. These methods propose localizing a vehicle relying only on visual information, since cameras are passive and inexpensive sensors that provide access to a rich information space.

The remainder of this section goes over the various concepts and implementations related to Visual Place Recognition (VPR), used in many autonomous vehicles to perform the localisation task: Section 2.1 presents the concept of VPR; Section 2.2 describes the traditional ways VPR is implemented in hardware; Section 2.3 describes the Log-Polar Max-Pi model (LPMP), a bio-inspired version of VPR.

### 2.1. Visual Place Recognition

*Visual Place Recognition* (VPR) is a field of research that addresses the issue of locating a place from visual information. The general idea is to determine the position at which an image was taken by comparing it with a geo-referenced database of images. The proposed methods have been used in many fields such as robotics [18,19], big data [20] and machine vision [21,22]. Each case has very specific constraints, notably in terms of computational time, accuracy and computational cost, which do not necessarily lend themselves to every use case [23,24].

From an architectural point of view, VPR models often follow the following workflow: first, an image is analysed to find its characteristic information; second, the detected information is transformed into a compact and meaningful location code; finally, the code is sent to a memory which has to store the location code (for the learning phase) or send information back (for the using phase), if the image belongs to an already known location [19,23]. Thus, it becomes possible to create a complete representation of an environment by memorising images at regular intervals.

In general, the performance of a VPR system is evaluated according to three criteria: first, the accuracy of the model, i.e., the average distance between the coordinates of an image to localise and the coordinates of the image that the model best recognises; second, the computation frequency of the model as a function of the number of images learned; and third the use of computing resources.

Currently, the best performing state-of-the-art models are deep models such as NetVlad, HybridNet or RegionVlad [25]. However, these models are very expensive in terms of computational resources (these models use a very large number of neurons to encode an image and often require the use of a graphics card) with higher complexity than traditional, non-Machine Learning approaches. For example, the CoHog model [22] gives a performance comparable to deep models while being much less greedy in computational resources. Nevertheless, this model is a VPR model of the “Global Handcrafted Feature” family and does not need to be trained before being used.

### 2.2. Hardware Implementation of a VPR Model

Although significant improvements have been made to VPR models, they are still not widely used as a main source of localisation for navigation. Performing a navigation task on an autonomous vehicle requires a fairly high localisation frequency that most VPR models cannot achieve. For instance, a vehicle must provide a location very fast when it is travelling in a motorway, so that it does not lose the navigation route. However, these constraints could be overcome by moving to a high performance implementation.

In this work, we propose to leverage a VPR algorithm by moving to a hardware implementation. We are particularly interested in the VPR models using neural networks. Neural architectures have the advantage of being particularly well suited to hardware implementation, in particular by massively parallelising the computations performed by neural networks.

We thus propose to leverage the Log-Polar Max-Pi model (LPMP) [23,24], a bio-inspired VPR model based on the use of a single conventional camera and neural networks based on a firing rate model. It has several interesting features for hardware implementation. First, LPMP is competitive with the state of the art, with high performance at small sampling scales. Second, it is a relatively simple model with a much smaller number of neurons than deep models. Third, although also based on a feedforward architecture, the model is not based on a succession of regular layers as in CNNs, but is rather composed of a succession of neuronal populations with different characteristics. In addition, Colomer et al.’s reference implementation is purely software-based (using Python). Without any optimisation, it cannot scale to satisfy an autonomous vehicle’s real-time constraints for localisation. It thus lends itself quite well to an FPGA implementation.

### 2.3. LPMP, A Bio-Inspired Model of Localisation

Among the various existing models, the Log-Polar Max-Pi model [6,24] (LPMP) depicted in Figure 1 represents our main research context for a hardware implementation of a visual localisation model (for further details, see Section 5). Inspired by the functioning of animal cognition, the model allows one to build a neuronal representation of an environment from visual information. In particular, it mimics a family of spatial neurons called place cells that can be observed in the hippocampal formation of mammals [26].

The model is used in two stages: The first is the learning stage, where a representation of the environment is learned in one-shot learning. During this stage, the model learns a number of images that are representative of a particular position. Depending of its version, the system learns the images, unsupervised and online, at regular intervals of distance or via a novelty detector (LPMP + vig). The second stage is the *query stage*. During this stage, the system analyses a batch of *N* images and returns their localisation within the learned representation. In the case of an autonomous vehicle application, the image analysed by the system is the one acquired via the camera.

Points of Interest detection

To localize an image, the model follows the classic VPR system pipeline (see Section 2.1): LPMP analyses an Image *I* to detect its points of interest (PoI).

Saliency points filtering

During this step, *I* is convolved with a Deriche filter, then with a DoG (Difference Of Gaussians) filter from which the most salient points are selected through a competition mechanism to retain only the most pertinent ones.

Points of Interest encoding

Then, LPMP encodes points of interest to obtain a compact and meaningful code of a place. At this point, the LPMP model builds a visuo-spatial pattern by performing a what–where merging in a Max-Pi layer, by using the visual identity of each PoI (via Log-Polar encoding) with their absolute orientation angle (obtained through azimuth computation and encoding). To encode the visual identity of a landmark, the LPMP model proceeds to a Log-Polar transform on the thumbnail around the PoI.

Memory querying

Finally, LPMP queries its memory to return the location that best matches *I*. To do so, the visuo-spatial pattern is passed to a neural memory: WTA (Winner-Takes-All). This memory contains the patterns of all previously learned locations (one location per neuron). At the time of a request, it returns by neuron a level of activity correlated to the proximity level of the current pattern and of all those learned. More details are provided by Colomer et al. [23].

Some Advantages and Limitations of LPMP

LPMP offers a lot of promises, in particular in terms of simplicity and consistency when the scene does not vary too much for a given lighting level. However, it is sensitive to radical light changes, e.g., going from a sunny environment into a very dark tunnel and vice versa. Yet, should the ambient light levels remain somewhat constant, LPMP is very accurate in terms of recognising a location even in the presence of human activity, resulting in moving objects (pedestrians, cyclists, other cars).

## 3. Computational Supports for Hardware Acceleration

Heterogeneous computing is a key strategy for meeting the requirements of many computation-intensive applications. However, current platforms which leverage both CPUs and FPGAs are commonly underutilised, as scheduling is often constrained to a run-to-completion model or to the acceleration of a single application at a time. With specifically designed hardware, reconfigurable fabrics represent the next possible solution to surpass GPUs in speed and energy efficiency. Various FPGA-based accelerator designs have been proposed with software and hardware optimisation techniques to achieve high speed and energy efficiency.

In contrast, GPUs offer up to 10 TOPs (i.e., *tera-operations* $\approx 10\times 10^{12}$ operations per second) peak performance and are good choices for high performance neural network applications. Development frameworks like Caffe [27] and Tensorflow [28] also offer easy-to-use interfaces which makes GPU the first choice for neural network acceleration. Alongside CPUs and GPUs, FPGAs are becoming a platform candidate to achieve energy efficient neural network processing. FPGAs can implement high parallelism and make use of the properties of neural network computation to remove additional logic. Algorithm studies also show that a Neural Network (NN) model can be simplified in a hardware-friendly fashion without hurting the model accuracy [29]. Therefore, FPGAs offer the potential to achieve higher energy efficiency compared to CPUs and GPUs. However, FPGAs are also infamous for being hard to design and program energy-efficient and highly performant neural networks, especially from a software developer’s perspective, compared to a CPU and/or GPU centric approach. Nevertheless, FPGAs are the best-suited devices if one requires end-to-end control, as they allow the systems engineer to design a tailored platform, provided the necessary field expertise is there to exploit them and yield a low power and energy footprint.

Several FPGA vendors propose AI-based or neural architecture-oriented solutions using FPGAs, including Xilinx products such as Alveo U55C [30], which is geared toward HPC and Big Data workloads, Versal ACAPs, a cloud-oriented platform, etc. However, such solutions rapidly become of limited use when considering “out-of-the-box” solutions, e.g., to implement bio-inspired algorithms in hardware, as resource utilisation can quickly lead to resource exhaustion when evaluating new algorithms. This leads the systems designer to explore other venues, such as multi-FPGA designs and solve additional issues, e.g., how to correctly synchronise FPGA systems connected through gigabit serial links and find the best communication schedule for a given (set of) design(s).

Hence, the reasons to resort to using a multi-FPGA board with an embedded high-speed interconnect are the following:Necessity of prototyping complex algorithms that need to be scaled.Leveraging Dynamic partial reconfiguration with the aim of reducing energy, power consumption and space locality task placement.Facilitating the incorporation of such middle-ware for partitioning: if there is a need to schedule work on multiple devices, how much workload should be executed on each device? For instance, scheduling 25% of the threads on CPU and 75% of the threads on FPGA.Leveraging high-speed transceiver protocols as an intrinsic property of FPGAs to communicate over multiple ones.

### 3.1. Gigabit Transceivers Interface

To bring out the high speed signals from inside the FPGA and interface with the real world, a needed demand for the use of transceivers is put in context. Compared to an approach using ordinary IO Pins for FPGA interconnection, it has several advantages: the provided bandwidth is very high while only few wires are required [31]. Thus, to leverage this FPGA’s feature aspect, a pre-developed hardcore IP has been incorporated within our FPGA ecosystem development.

In our work, we use the LogiCORE IP 7 Series FPGAs Transceivers Wizard, which is able to automate the task of creating HDL wrappers to configure on-chip FPGA transceivers. Wizarde’s customisation GUI allows users to configure one or more high-speed serial transceivers using either pre-defined templates supporting popular industry standards or building a protocol from scratch [32].

An interconnect framework for FPGAs based on multi gigabit transceivers (MGTs), typically available in modern reconfigurable devices, is proposed by Dreschmannetal et al. [31]. The framework provides higher bandwidth while using fewer pins, compared to existing approaches based on ordinary FPGA IO pins. Unlike other implementations using MGTs for device interconnection, special care has been taken to achieve high throughput and data integrity while keeping latency, resource usage and protocol overhead very low. Depending on the available MGTs, the bandwidth per connection reaches from 3.125 to 28 GBit/s, allowing large amounts of data to be moved quickly between multiple FPGAs [31].

Yangfane et al. present in their work a network on chip (NoC) emulation at the physical level [33], with two levels of interconnects that are adopted to mimic NoC on-chip communications: high bandwidth and low latency parallel on-board wires and high-speed serial multi-gigabit transceivers, which is particularly important, as it helps the proposed NoC emulation platform to scale well as the NoC size increases.

Aloisio et al. show that high-speed serial links are a key component of data acquisition systems for high energy physics [34]. They carry physics event data and often also clock, trigger and fast control signals. The authors demonstrated that the jitter on the clock recovered from the serial stream is a critical parameter, since it directly affects the timing performance of data acquisition and trigger systems. While FPGAs include multi-gigabit serial transceivers, which are configurable with various options and support many sorts of data encoding.

### 3.2. Difference of Gaussian IP for Feature Extraction

The Intellectual Property (IP) implemented by Fiack et al. [8] resorts to several types of operations, including gradients, as well as several differences of Gaussians (DoGs) operations. It provides pixels data of each landmark identified on the captured image, based on a sequence of difference of Gaussians. DoGs are used in multi-resolution methods to avoid expensive computations due to filtering operations. The algorithm used to construct the processing phases of each level of resolution is detailed and evaluated by Fiack et al. [8] on FPGAs. Their IP is based on successive image filtering operations with 2D Gaussian kernels. It detects features in an image stream and then passes them to central core as post-processing.

## 4. Modelling the Bio-Inspired Algorithm for FPGAs

N-LOC implements a bio-inspired neural architecture, which performs visual localisation by mimicking the functioning of the mammalian brain [6]. To localise an image, the architecture encodes landmarks (given by the image-processing IP) in a unique visuo-spatial pattern via several neural structures. Our resulting IP is composed of three stages: (1) computation of the landmark visual signature via a winner-takes-all network (WTA), stored in the *signature layer* (SL), along with the computation of their angular position in the *azimuth layer* (AL); (2) merging of SL with AL via a *Spatial Working Memory* (SWM); (3) computation of “place cell” (PC) activity via winner-takes-all for an appropriate localisation.

### 4.1. Visual Signature Computation

The computation of the visual signature landmarks relies on a *winner-takes-all* group of neurons (WTA). It consists of a neural network, carrying out input signal discrimination through competition. This WTA models cognitive properties, e.g., decision-making, visual and auditory attention and selective amplification. A WTA consists of a weighted average-based computation, with a post tree-reduction selection, in order to only keep the highest activity among activated neurons in that group of neurons, as shown in Equation (Equation 1):(1)$$S_{i}=1-\sum_{j=1}^{N_{pixels}}\left(\right|E_{j}-W_{ij}\left|\right)/N_{pixels},$$
where *i* is the index of the neuron being considered, *j* is the index of the pixel being processed, $W_{ij}$ is the weight of the *i*-th neuron processing the *j*-th pixel of the Log-Polar thumbnail (centered on a selected point of interest) coding a visual landmark and $N_{pixels}$ is the total number of pixels per landmark thumbnail.

### 4.2. Angular Position Computation

The computation of the landmark angular position $\theta_{l}^{north}$ (or azimuth) is described in Equation (Equation 2), where *l* is the $l^{th}$ neuron in the Azimuth-Layer (AL) vector and relies on the interpolation between the landmark angular position $\theta_{poi}^{ego}$ and the vehicle orientation $\theta_{Vehicle}^{north}$ (each neuron encodes 2${}^{\circ }$; we need a population of 180 neurons for 360${}^{\circ }$ of total camera angle). Equation (Equation 2) yields:(2)$$\begin{array}{c} \theta_{l}^{north}=\theta_{poi}^{ego}+\theta_{Vehicle}^{north}\;\left(\mathrm{mod}\;2\pi \right)\;, \end{array}$$
where $\theta_{l}^{north}$ is in radians. This information is encoded in the form of a population of neurons, a bio-inspired neural structure which encodes the current azimuth value in the form of an activity bubble.

### 4.3. Spatial Working Memory

The Spatial Working Memory (SWM) is an $N_{S}\times N_{A}$ pixel matrix. $N_{S}$ is the number of neurons in the SL vector, and $N_{A}$ is the number of angles considered in the model. $N_{A}$ is in effect a subsample taken from the Azimuth Layer AL. This is illustrated in Figure 2, $N_{A}=4$, where each angle is equal to $45^{\circ }$. In our actual experiments (see Section 5), we set $N_{A}=3$. We denote $N_{\mathrm{SWM}}$ the total number of values which compose the SWM. *A* is the number of subsamples in AL, with A=4 (i.e., each group holds 60 neurons). Equation (Equation 3) provides the potential of the $il^{th}$ neuron and Equation (Equation 4) yields its final activity, where *f* is the activation function of sigmoid, which is applied to normalise the final results:(3)$$I_{il}\left(t\right)=\left(s_{i}\left(t\right)\cdot w_{i,il}\left(t\right)\right)\cdot \left(\max_{j\in N_{a}l}\left(a_{j}\left(t\right)\cdot w_{j,il}\left(t\right)\right)\right),\;\mathrm{and}$$
(4)$$x_{il}\left(t\right)=f\left(x_{il}\left(t-1\right)+I_{il}\left(t\right)-x_{il}\left(t-1\right)\cdot In\left(t\right)\right).$$

The connection weights between AL and SL are assigned to 1 in the SWM. The execution workflow of the second WTA is described in Figure 2.

### 4.4. The Place Cell Neuron Group

The activity of the SWM matrix characterises the current location. It is memorised in a *place cell vector* (PC) of $N_{P}$ neurons, which is then fed to another WTA process to select the most active neuron. It then models “place cell neurons” found in the hippocampus that have a close activity [35]. $N_{P}$ refers to the maximal number of images that the N-LOC IP can memorise. Equation (Equation 5) gives the activity of the neuron *i* at time *t*. Each neuron in PC holds connections related to learned images: the activity of one neuron gives the similarity between a learned place and an image. Thus, it can provide the appropriate information about the current location to the localisation system:(5)$$P_{i}\left(t\right)=1-\sum_{j=1}^{N_{\mathrm{SWM}}}\left(\right|W_{ij}^{\mathrm{SWM}}-x_{j}\left(t\right)\left|\right)/N_{\mathrm{SWM}}.$$

### 4.5. Modes of Operation

The N-LOC architecture has two modes of operation. The first is the *learning* mode, where the connection weights of its different components can be updated to memorise new images. It is triggered when the autonomous car starts the localisation process or when the car enters a new location. The second is the *using* mode, where the connection weights are fixed. This is where the actual evaluation of the localisation performance per each captured image throughout the camera occurs.

## 5. Hardware Implementation on the Wizarde Platform

This section intends to motivate and demonstrate why FPGAs were used as accelerators to implement LPMP in this work: they tend to offer more throughput with a lower footprint than (embedded) GPU-based systems, but also there are platforms geared toward prototyping which can help with our future designs for distributed localisation tasks.

### 5.1. Accelerating the Localisation Task: FPGA vs. GPU

The (LPMP) localisation task requires autonomous vehicles to process visual information in hard real time to feed the Spatial Working Memory and ultimately efficiently compute the vehicle’s location. Quite simply put, the higher the speed of the vehicle, the higher the processed frame-rate must be in order to allow the decision-making system to perform safely and efficiently. Hence, decision-making relies on a very low-latency localisation task, especially at high speeds.

Table 1 compares a relatively high-end CPU+FPGA System-on-Chip (SoC), a Xilinx zc706 board, with two different GPU-based SoC embedded platforms: Nvidia Jetson TX2 and Jetson Xavier. The table displays the host-accelerator latency overhead for data transfer roundtrips of very small data packets (a single 32-bit word at a time, which is representative of what our system must deal with, with an objective of very low latency). As can be seen from Table 1, while the Jetson Xavier system yields a much lower latency than the Jetson TX2, a 32-bit host-GPU roundtrip is still ≈100 times higher latency-wise than its SoC-FPGA equivalent. This confirms that data transfers between host and accelerator tend to favour FPGA-based systems: several studies, e.g., Qasaimehe et al. [36], observe between 100× and 10,000× shorter latency or higher bandwidth when comparing CPU + FPGA and CPU+GPU (depending on which kind of Nvidia platform is used). Likewise, power consumption is largely in favour of reconfigurable systems when compared to GPUs, as shown in Section 6, which is in line with what Qasaimeh et al. detail in their work [36] (see Section 7 for more information).

In addition, as we explain in Section 5, the power consumed by N-LOC is around ≈3 W for the whole system, whereas in the case of Jetson systems it is situated in the 7.5–15 W range, i.e., a Jetson system would consume between two and five times more power than our resulting hardware implementation of LPMP. Thus, this justifies our choice of using FPGAs as a platform to prototype and implement the bio-inspired neural architecture.

### 5.2. Hardware Implementation on a Multi-FPGA Platform

Wizarde is a unique, custom-designed multi-FPGA board, set up as a 3×3 mesh of system-on-chip (SoC) tiles. Each tile features a Zynq xc7z045ffg900-2 SoC (based on the Zynq-7045 chip, as with the zc706 board), which embeds a dual-core Cortex A9 processor and a Kintex-7 FPGA (Table 2 describes a single tile) [37]. Wizarde enables us to envision various scenarios, e.g., a multi-camera processing phase, where each tile is tied to a single camera and computes its own image processing phase to feed a shared bio-inspired neural architecture.

An important feature of Wizarde is the gigabit transceiver interface set between two neighbouring tiles. This will allow hardware tasks, i.e., their bitstream representation, to be mapped to different modules of the target FPGA(s) depending on the run-time context. For example, due to specific resource contention, a given reconfigurable region may not be available to a task which used to run on it. As a result, such a task may be run on a different available slot somewhere else in Wizarde.

Wizarde offers the following advantages from a research viewpoint:Unique, custom platform;Intent: help prototype complex applications, possibly requiring multiple SoCs/FPGAs;3×3 2D tile mesh.

In addition, each tile is independent (equipped with USB, Ethernet, micro-SD, DRAM, etc.), but all tiles are connected to their neighbours through gigabit transceivers, as shown in Figure 3.

The system is meant to retrieve landmark data resulting from the image processing carried out via the VITA-2000 camera module [38]. The organisation of the architecture is depicted in Figure 4. It is composed of pipelined blocks of custom IPs, which take their input from a camera through a streaming interface. The image processing IP can be configured in software by the CPU part via a memory-mapped interface, i.e., a Xilinx AXI bus, which bridges the Processing System (PS) and the Programmable Logic (PL) of the SoC. The results can then be processed in software as follows:A differences of Gaussian (DoG) or processed intermediate image, selectable thanks to a dedicated register.The list of pixel data extracted and sorted by the IP at the different frequency bands and the list of Log-Polar features associated with each keypoint, i.e., a set of computed pixels. They are gathered into *landmarks*. Each landmark contains 12×12=144 pixels.

**Figure 4 sensors-23-04631-f004:**
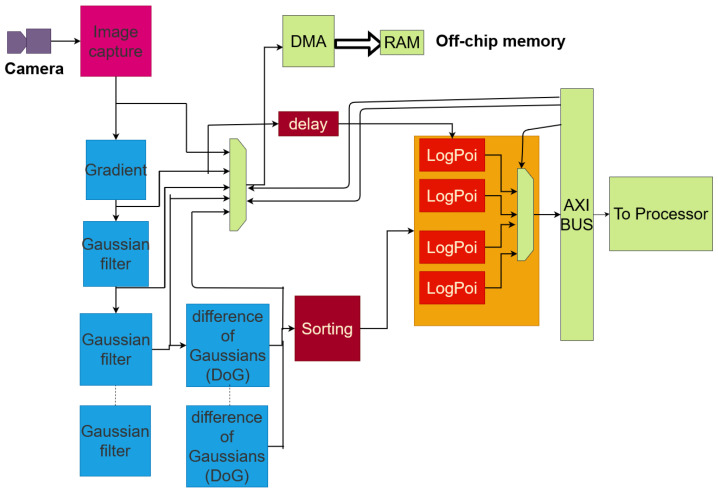
Overview of the scaling-based image-processing IP.

The detected features are identified by their coordinates. The pipeline follows a pixel dataflow model of computation. The coordinates of the pixels are mandatory for each IP and are read in order to ensure that the memory mapping of buffers is adequate and to avoid side-effects.

Moreover, the learning phase of the visual cells for localisation needs to know the coordinates of the pixels that each landmark contains. The pixel coordinates must then be transferred from the image processing IP to the bio-inspired IP. The pipeline allows these data to be either delayed or regenerated by each processing IP depending on its latency.

The total power consumption (i.e., the total on-chip power of the design), measured via Vivado, yields 2.981 W. Table 3 shows the resource usage on the Wizarde platform.

The software memory footprint is rather low: each neuron carries an 8-bit value. If we consider 4000 neurons in total and include the Python interpreter code size to run the program, the resulting memory footprint remains below 1MiB. Running such a program on a high-end microprocessor (e.g., Intel Core i5, i7, …or AMD Ryzen family) will easily have both the code and data fit in its L2 or L3 cache. Such a low memory footprint makes it rather inadequate to run on a GPU, as the PCIe bus latency would likely kill most of the performance gained.

### 5.3. Fixed-Point Arithmetic

As is traditional when targeting reconfigurable hardware, fixed-point values are used to represent decimal values. Our implementation uses 8-bit values, with 2 bits for the integer part and 6 bits for the fractional one. This is because all neurons’ weights and values are between 0 and 1. We have experimented with different resolutions, i.e., 8, 10, 12 and 16-bit fixed-point numbers. Our experiments showed no significant improvement in accuracy for place cell activation.

### 5.4. Towards Using GTX Transceivers for a Data-Transmission over Wizarde Platform

As seen in Figure 3, the Wizarde platform targeted for our experiments features gigabit transceivers (GTX). They will be an essential tool to deploy our N-LOC IPs across multiple tiles and have them communicate with each other.

The data in the frame generator (stored in BRAM) are sampled with a reference clock ref_clk of 50 Mhz before being sent to the north-west tile (NW) via GTX. The burst sending is expected with frequency of 3.125 Ghz, as shown in Figure 5.

The different signals to be considered in our GTX transmission protocol are as follows:Error_count: it should be NULL based on chosen frequency.Dynamic reconfiguration port (DRP) clock.Rx_reset_done: should be set to 1 when data are correctly received.

### 5.5. Experimental Results

This section presents the results of our implementation: resource utilisation, latency and throughput performance, using a single N-LOC IP block over one FPGA tile. This first study represents the starting point for the scalability of the approach.

#### 5.5.1. Experimental Setup and Implementation Parameters

The target hardware platform is based on the Zynq-7045 system-on-chip, which features a dual-core ARM Cortex A9 processor, coupled with a Kintex-7 FPGA (see Table 2 for additional details). The Processing System (PS, featuring the dual-core Cortex A9) runs a bare-metal executable which streams pixels to the programmable logic (PL). N-LOC was designed using Vivado HLS, i.e., using the OpenCL language with FPGA-specific *pragmas* to specify bus sizes, etc. The PL implements our N-LOC IP, which was synthesised and implemented on the FPGA part of the Zynq SoC. Hence the performance results shown below are obtained using an actual implementation of our design (not a simulation). Moreover, we used Vivado 2016.4 to synthesize the IP on our FPGA. The Wizarde board was validated with this toolchain and we have not yet qualified and validated Wizarde with a more recent Vivado version (in our future work, we will generate a device tree for a Linux distribution which is compatible with a more recent version of Vivado). Resource-wise, we compared the synthesis reports of Vivado 2016.4 and 2019.3. The difference is below 4%. While the synthesis process of a more recent version of Vivado may yield better resource usage in general, we believe the nature of our IPs would see only marginal gains compared to Vivado 2016.4 in our specific case. Moreover, the number of DSPs would remain the same, even in parts of the IP which can take advantage of them (e.g., the WTA part).

Ideally, we would like to plug the image-processing IP (which performs the pyramidal decomposition of the acquired images) directly to N-LOC. However, in order to correctly isolate the N-LOC part of the processing chain, we opted to use a small bare-metal executable which will stream images taken from the Oxford dataset [39]. This will also prove useful when we evaluate the distributed N-LOC architecture, as it also requires a minimal runtime system to orchestrate data synchronisation between multiple N-LOC instances (see Section 6).

Hence, a set of parameters is given to obtain around 90% accuracy when comparing the value of computed place cells after the second WTA and the value in pre-learned images. As explained in Section 5.3, accuracy does not change significantly when increasing the fixed-point number format to a higher number of bits. The initial training/learning phase uses as many images as there are place cell neurons. Our bio-inspired IP uses the following parameters:On the target Zynq-7045 chip, the signature layer (SL) can fit at most 1600 neurons; the SWM 4800 neurons; the place cells vector 100 neurons, with 16 landmarks per image.The azimuth layer (which represents the angle orientation of each captured image) contains 180 neurons.The values of azimuth neurons are fixed for this experimentation; varying azimuths will be added in the future.We parameterised N-LOC with three configurations: 30, 60 and 90 place cell neurons. They represent the number of images that need to be learned in the localisation process.We used 100 images to “feed” N-LOC.

We compare our bio-inspired neural IP with Colomer’s Python-based application [23] to build the place cells vector. As will be detailed in Section 5.5.2, we considered two versions of the software implementation: the baseline and its optimised version. Both make use of Cython, numpy and OpenCV to speed up computations. However, the optimised version of the software implementation is also parallelised and makes better use of Cython to help guide it toward more efficient code generation. As detailed in Section 2, image pre-processing is performed either using OpenCV in the Python script or in the image processing IP. A previous version of this model was validated on large datasets, as shown by Espada et al. [40]. Iterations over LPMP led to an optimised code, which was tested in a mobile robot in a real-life environment (closed tracks) to evaluate the performance of a software LPMP implementation. As Colomer et al. show [23,24], the algorithm is accurate and correctly identifies landmarks. However, this implementation does not go fast enough to scale at higher speeds. In addition, only the *learning* and *using* phases leading to building the place cell vectors are evaluated; i.e., we do not time the video pre-processing which does the gray-scaling and applies difference of Gaussians and Log-Polar conversion performed by the previous component in the pipeline. As we described above, the reference code is written in Cython, with uses of NumPy and OpenCV where appropriate. Our application directly embeds all (grayscaled) images to exploit and as a result we subtracted the time taken by the Python application to perform all the processing prior to the actual *learning*/*using* phases, i.e., I/O operations to read image files, putting colour images to gray scale, dividing images into landmarks, etc. These operations represent roughly 20% of the total execution time in the baseline and optimised software implementations. We used different learning configurations: the system had to learn using 30, 60 and 90 gray-scale images taken from the Oxford dataset [39] to train the system (*learning* mode), then used 100 images in total in the *using* mode. Each image has a resolution of 640×400 pixels. Python results were obtained with Nvidia Jetson TX2 and followed the same principles for the learning/using ratio.

Colomer et al. already showed how the LPMP model behaves with large datasets [23]. In particular, real-life environment data obtained on close tracks using a mobile robot running the software version of LPMP were gathered and validated the LPMP model’s accuracy and precision [41]. The (optimised) reference code correctly identifies the right landmarks in a real-life context. Moreover, our hardware implementation also selected the same landmarks in our experiments as the software reference implementations. Hence, to compare our hardware implementation to the software implementation used by Colomer, we consider it is sufficient to resort to the Oxford dataset and run it on both software and hardware implementations to compare the two and check how faithful N-LOC is to the original software implementation.

#### 5.5.2. Resource Consumption and Performance Gains

This implementation requires ≈26% of available LUTs and ≈50% of BRAM blocks. The amount of BRAM used is consistent with the format used for the SWM, which is essentially a dense 1600×3 matrix of neurons yielding 8-bit values. Table 4 provides details for specific resources required to implement our neural IP to process 8 or 16 landmarks, respectively. We cannot consume more resources to increase the number of neurons because space must be made to also fit the image processing IP.

Result numbers were obtained using Vivado’s HLS framework v2016.4, coupled with a custom workbench. The design was implemented on a ZC706 board, which features the same Zynq-7045 SoC that Wizarde uses. This board allowed us to obtain performance measurements, resource consumption and power consumption estimates. We used the parameters described in Section 4. Further, we also used those parameters to implement N-LOC.

We conducted our experimentation using three different *learning* configurations, to compare both software and hardware approaches. We set the number of place cell neurons at 30, 60 and 90, respectively, and then test and evaluate the *using* phase on 100 images. Our FPGA-based implementation outperforms the baseline Python-based reference implementation [23], with an average throughput of 52 images vs. 7 images. Moreover, the LPMP algorithm performs 50961 operations per image for 16 landmarks of one processed image. Thus, by multiplying those values, we obtain 0.032 GFLOPS/s for the Python implementation and 2.3 GFLOP/s for our FPGA-based solution. The learning phase latency is 6× shorter and for the using phase, the latency is 9× shorter, with a total throughput ≈7× larger. See Table 5 for more details about the performance comparison.

A traditional Python implementation is around 10 times slower than sequential implementation with close-to-the-metal languages such as C. The first implementation of Colomer et al. is not pure Python: it resorts to Cython, which translates Python code into C and compiles it natively and also makes use of numpy and OpenCV, which are written in C and C++. In their second implementation, processing neurons in SL, SWM and PC structures are parallelised in the *using* mode. As a result, it is much faster than the sequential reference implementation we use as our baseline. In terms of *learning* latency, the optimised LPMP reference improves significantly its processing performance compared to the baseline. N-LOC still yields better throughput and latency compared to the optimised version (≈9× lower). The optimised version stores all of the image pixels in one shot, whereas we stream them one by one. However, while the optimised version does perform twice as well as the baseline for the *using* latency, NLOC still outperforms this optimised version, as its latency is ≈2–3× lower (≈9× compared to the baseline). The *learning* phase is dominated by data (pixel) transfers, whereas the *using* phase is significantly more intensive computationally speaking. Thus, as the number of image comparisons grows in the *learning* phase, the *using* phase takes progressively longer in time [23]. Our implementation of the *using* phase’s latency is ≈9× lower than the baseline and ≈2–3× lower than the optimised version. Consequently, the throughput of the optimised LPMP version is ≈1.5–2× faster than the baseline when considering total throughput, but N-LOC’s total throughput is itself ≈3–4× higher than the optimised LPMP version (and ≈7× higher than the baseline). All the details of our experiments are shown in Table 5. Each image is composed of 16 landmarks, of resolution 12×12 pixels.

Figure 6 illustrates the increase in the number of resources used in the FPGA for each experimentation in which we varied the number of neurons in place cells for each N-LOC IP during the design implementation using Vivado. Our analysis revealed that the LUT and BRAM were the primary resources consumed. This can be attributed to the N-LOC architecture’s reliance on both storing pixel values on neuron weights and performing simple mathematical operations on LUTs rather than using DSP.

Table 4 illustrates the various metrics we considered. Multiple experiments allowed us to assess the accuracy of our visual recognition process compared to pre-learned images following Colomer’s model. It is predicated on the number of landmarks per image. Table 6 shows resource utilisation of the overall application implementation. Hence, a resource consumption trade-off must be considered to make the design fit the board.

Table 7 details the power footprint of the whole hardware-based application (image processing and neural IPs). The overall footprint (image processing IP and N-LOC IP) consumes 2.75W. This is to be compared to the nominal power consumption of Nvidia Jetson TX2 used for our experiments (with a power consumption of 7.5 W–15 W).

## 6. A Distributed N-LOC Architecture

Depending on the requirements of the target localisation application, it may be necessary to implement a larger neural network, which does not fit onto a single FPGA board. For example, the original bio-inspired algorithm proposed by Espada et al. [6] requires ≈10,000 neurons in total, whereas our experiments on a single FPGA board can fit at most ≈4600 neurons on the target zc706 board. As a result, it is necessary to explore the possibility of producing a *distributed* version of N-LOC, which could scale as needed. By expanding the neural architecture of the localisation task, the autonomous vehicle’s capacity to localise itself mechanically increases. For instance, a larger neural network implies a larger pre-learning environment as well—and a larger on-line learning capacity in general (see Section 4 for details about how the bio-inspired algorithm implemented in N-LOC works). Hence, a large neural architecture results in a wider and more efficient place-recognition task, as shown in Figure 7.

### 6.1. Distributed N-LOC: Principles

#### 6.1.1. Distributed Learning and Using Phases

Learning phase

In a distributed N-LOC environment, the *learning* phase workflow is as follows:N-LOC IPs are duplicated and distributed on different tiles.The pixel stream is connected to the appropriate N-LOC block if its neurons’ weights are not saturated.If an N-LOC block is saturated during the learning phase (i.e., the maximum number of neurons to initialize has been reached), we switch to the next available N-LOC block, to carry on the ongoing or further learning of different captured images.The master controller is in charge of communicating with all N-LOCs, using a bidirectional communication protocol.

Using phase

Likewise, the *using* phase workflow works as follows:The pixel stream is connected through all N-LOC blocks simultaneously.All N-LOC blocks *simultaneously* perform different computations based on different pre-learned information.A threshold-based comparison is set by the master controller, to select the highest activated neurons among the N-LOC blocks.

A proof of concept of this distributed architecture was implemented on a single Zynq-7045’s programmable logic, with 3 N-LOC block instances. We next discuss the possibility of using gigabit transceivers to enable fast communications between GTX users.

#### 6.1.2. Experiments with Distributed N-LOC on a Single FPGA

Experimental setup

We tested our distributed N-LOC model by instantiating three N-LOC blocks on a single Zynq-7045 SoC. The controller is implemented in software on a bare-metal ARM Cortex A9 microprocessor. The FPGA and Cortex A9 are linked through an AXI lite bus.

Our distributed N-LOC system is composed of 3 × N-LOC instances of 30 place cell neurons each, for a total of 90 neurons. Hence the total number of neurons in these three place cells is 90, which is 10% smaller than the initial single-block N-LOC instance studied in Section 5.

As described earlier, we compare our results with a Python program which makes use of NumPy, OpenCV and is compiled with Cython. The experimental conditions are the same as described in Section 5.5.

Experimental results

Table 8 shows the latency and throughput calculated from both the learning phase (where a single N-LOC is active at a time) and the using phase (where all N-LOCs compute in parallel) are assessed at the same time. The latency performing one learning phase is roughly the same, whether the N-LOC system is distributed or not. The using phase fares slightly better with a distributed N-LOC system when processing a single image. In both cases the global throughput to both learn images and use this knowledge to localise the vehicle is an order of magnitude better than with the reference application.

Moreover, while the learning phase only copies pixel values as weight into neurons, the using phase performs much more computationally intensive operations, as there is a winner-takes-all (WTA) stage to update the signature layer (SL), then an update of the Spatial Working Memory (SWM) and once the image has been fully processing, i.e., in our case, once all sixteen landmarks which compose an image have been processed, yet another WTA operation takes place to select the most active place cell neuron and decide if the measured score is high enough (i.e., has reached the preset value threshold). Hence, the using phase latency is bound to be much higher than the learning one.

Table 9 provides speedups of a monolithic and a distributed N-LOC system vs. the optimised software implementation. Compared to the optimised reference application, N-LOC is 4–6× faster, but compared to the individually measured learning and using latency, this performance is rather low. It is important to note that in the reference code, as the number of place cell neurons increases, the processing time also increases dramatically: the learning latency reported in Table 5 is 60% higher than the 1×90 configuration shown in Table 8; the using latency is 16% higher; and the total image throughput is 7% lower. While we must make use of additional FPGA units to extend the size of our network, the intrinsic parallelism used in the various phases ensures that image processing latency remains relatively constant; the only true bottleneck is the communication between the processing system (PS) and the programmable logic (PL).

In general, the main bottleneck in the N-LOC hardware implementation is the naïve implementation we carried out, where we isolated the N-LOC instances as much as possible, but which results in multiple AXI-Lite roundtrips between the processing system (PS) and the programmable logic (PL). A more involved architecture would have the PS only send messages once for broadcasting, with a hardware-based broadcasting designed internally to carry the data frames to each N-LOC instance. However, this approach also has drawbacks: it makes the overall architecture more “rigid”, which in turn may hamper the capacity of the system to scale with several N-LOC instances, e.g., up to eight or even nine, if we target the Wizarde platform. Further, one of the inherent difficulties dealing with FPGAs stems from inherent issues related to (reconfigurable) hardware and the use of HLS: as we grow from 60 to 90 place cell configuration, in order to maintain acceptable timings and clock distribution within the system, we must reduce the clock frequency from 100 MHz to about 70 MHz. This is a limitation tied to a relatively naïve approach in our own design and we plan on exploring ways to increase the clock frequency to improve performance.

Power-wise, both the PS and PL parts of the target Zynq SoC see a slight power consumption increase, as shown in Table 10. This is not unexpected: on top of sending pixels to the FPGA, the Cortex A9 core is now also tasked with selecting N-LOC instances during the learning phase, but also sending the pixel stream to all instances during the using phase. Likewise, each N-LOC instance requires proportionally more FPGA resources compared to their single N-LOC counterpart. The resulting total power consumption (static + dynamic) is around 2.8 W, with ≈0.2 W for the hardware part. Compared to the Nvidia Jetson TX2 board used to run the reference program; this is a 5.5–6× improvement.

Table 11 shows the resource utilisation of the overall application implementation. The 1×90 N-LOC instance requires fewer resources than its 3×30 N-LOC counterpart: it requires 20% fewer LUTs, 63% fewer flip-flops and 65% fewer DSPs. However, the relatively large dense memory matrix required by a monolithic 90 place cell neuron network requires a complex BRAM usage via synthesizer and BRAM usage is twice as large as with 3×30 neurons. For the DSP part, this is a limitation tied to the need for computations of a single IP: with a single 1×30 N-LOC block, we reach almost the same amount of DSPs used as with 1×90.

Table 9 and Table 10 summarize all the synthesis results generated and presented within our works, along with some ratio comparisons in term of latency, throughput and power consumption.

Table 12 provides the performance per Watt of several configurations, one with 100 place cell neurons and the other with 90 neurons. The performance ratio with power consumption when comparing the reference code with N-LOC instances varies from 4× to 7× (for 30, 60 and 90 place cell neurons according to the results shown in Table 5). This metric is obtained by computing the following: (1) Each image is partitioned into 16 landmarks, each composed of 12×12 pixels; i.e., there are 2304 pixels to process in each image. (2) During the *using* phase, there are three distinct types of operations: For the first winner-takes-all, each pixel is broadcast to each element of the vector of neurons. To process all pixels for a single neuron, there are [(12×12)×16]×3=6912 operations to perform. Since there are 16×100=1600 neurons in the signature layer, the total number of operations to perform in the neuron vector is 6912 × 1600 = 11,059,200 operations. Once this is done, there is a MAX computation to be performed between all 1600 neurons, i.e., 1601 additional operations; to update the Spatial Working Memory, a single neuron is updated according to all azimuth neurons for the image orientation, resulting in 362 operations for that step, and there are 101 max operations to execute to perform the second winner-takes-all. (3) We sum all operations required to perform both WTAs and the SWM update, which yields $\approx 11.1\times 10^{6}$ operations = 11.1 MOPs. To obtain the performance per Watt, we compute the following, $Perf\_per\_Watt=\frac{N_{\mathrm{ops}}}{Power}$, using the throughput values reported in Table 5 and Table 8, as well as the power consumption of Table 7 and Table 10.

Hence, the first winner-takes-all step is overwhelmingly more computationally intensive than the other steps. The total number of operations to process a single image is ≈11.1 $\times 10^{6}$ operations or 11.1 MOPs. The reference code runs on Nvidia Jetson TX2 with a thermal design power (TDP) up to 15 W, which we used as the baseline to compute the performance per Watt of various configurations. As the table shows, there is a 4× to 6× ratio in favour of our N-LOC design.

### 6.2. Communication Protocol via GTX Transceivers

To bring out the high-speed signals from inside the FPGA and interface with the real world, a needed demand for the use of transceivers is put in context. Compared to an approach using ordinary IO Pins for FPGA interconnection, it has several advantages: the provided bandwidth is very high while only a few wires are required [31]. Thus, to leverage this FPGA’s features aspect, a pre-developed hardcore IP was incorporated within our FPGA ecosystem development. The LogiCORE IP 7 Series FPGAs Transceivers Wizard is a type of serial communication that will be used and is already incorporated in the Wizarde board; it provides the ability to automate the task of creating HDL wrappers to configure on-chip FPGA transceivers. The Wizard’s customisation GUI allows users to configure one or more high-speed serial transceivers using either pre-defined templates supporting popular industry standards or building a protocol from scratch [32].

#### 6.2.1. Highspeed Transceivers on the Wizarde Platform

As seen in Section 2, the eventual platform on which to run N-LOC is Wizarde, a 3×3 tile board, with a 2D mesh communication network composed of gigabit transceivers (GTX). Hence, we must define a protocol and a communication scheduling policy to leverage the GTX. The data transfer policy relies on streaming pixels at each rising edge into our N-LOC blocks (when the data are sent to the SL layer). Then, all the information required by each block is sent to it accordingly. We evaluated that 32 bits is the maximum packet size required to transfer data for both RX and TX sides of the GTX. Our design is illustrated in the upper-left corner of Figure 8, including the various required word sizes for TX and RX.

#### 6.2.2. GTX Micro-Benchmarking in Wizarde

We use Aurora, a LogiCORE IP [42] designed to enable easy implementation of Xilinx transceivers while providing a light-weight user interface on top of which we can build our own protocol. This IP offers sufficiently low overhead for our needs and will allow us to build our own higher level protocols in the future while maintaining a high scalability potential.

Specifically, we leverage an 8B/10B encoding, a protocol for a high-speed serial data transmission. It provides a good clock recovery on reception and balances the number of zeros and ones to avoid the presence of a direct-current (DC) on the line. It is used in some versions of Ethernet-based network links.

The Aurora IP exposes an interface with an AXI4-stream bus, which will allow us to send high speed data, e.g., via its external DDR memory and a DMA, from the processing system part of the Zynq to its programmable logic part.

We implemented tests to validate that tile-to-tile data transfers are indeed correct on the Wizarde platform. We specifically targeted communications between the north and north-west tiles. The benchmarks are carried out at 3.125 Gbps and 6.25 Gbps, as the maximum admissible frequency for the Aurora 8B/10B IP is 6.6 Gbps. The Aurora configuration is shown in Table 13 for 6.6 Gbps.

We use the IP in simplex mode. The north tile will be in the transmit mode while the north-west one will be in the receive mode. Our clock reference on Wizarde is set to 125 MHz, to be able to boost the frequency up to 6.25 Gbps. Each tile has a pair of MGT links connected to its nearest neighbours (e.g., the central module has four pairs of MGTs to provide a high-speed transmission to each of its immediate neighbours).

Finally, to carry out our tests we used the example design with the dedicated core IP, which has modules for frame-generation (on the TX side) and frame-verification (on the RX side). The frames are composed of pseudo-random numbers sent in the AXI4-stream format.

Our tests show the data we send (TX) are identical to the received data (RX), with a delay overhead of (≈clk_cycl/10). As shown in Figure 9, we send arbitrary fixed numbers from side to side and then evaluate the received data (registered in BRAM memory), according to the transmitted ones on the TX register.

To ensure a proper activation and reset of the GTXs on both sides of the transmission, we added some functionalities according to Xilinx [42]; for more documentation on the chronogram and the VHDL codes used for both sides of TX and RX and for a more efficient control, see page 55 [42].

Figure 10 and Figure 11 showcase the resource utilisation for post-implementation, using two different boosting clock frequencies 3.125 Gbps and 6.25 Gbps. The illustrated results are generated and exposed from both side of tiles on north and north-west.

### 6.3. Toward a Distributed N-LOC Architecture on Wizarde

This section discusses the possibility to implement a distributed N-LOC architecture using a multi-FPGA platform. We will use Wizarde (See Section 2) as our target. Our reasons to resort to Wizarde are threefold: (1) Beyond the intrinsic overhead induced by a distributed architecture and its associated control signals, one of the reasons our first attempt at implementing distributed N-LOC does not perform as well as a single N-LOC block is the very small size of each neural network, which can be alleviated if a sizeable portion of each tile involved in the design can be leveraged (as one tile is roughly able to store 4600 neurons with 100 place cell neurons); (2) the GTX links coupled with the AXI Stream protocol should offer a more asynchronous way of transferring data between the controller (still implemented in software) and its neighbouring tiles, which should reduce communication overheads (for both throughput and latency); and (3) this is the only way we can eventually implement a large neural network—large enough to be useful in a self-driving car. Moreover, such a network could be grown dynamically and on demand, according to the computational needs of the current context in which the car is situated.

#### 6.3.1. Changes to the Original Architecture

We target four tiles in the Wizarde board: for instance, the central tile, as well as the north, east and west tiles. The latter tiles implement an instance of the N-LOC IP, combined with a GTX interface (see Figure 8). The central tile implements the image processing IP (DoG) and orchestrates communications across all tiles via the GTX interface. The communication scheduler is implemented in software on the central tile, using the ARM Cortex A9 processor.

More details of N-LOC data exchange of buffer size are given below:The compass value (image orientation, i.e., azimuth values) is sent once for each image to process. Azimuth values are computed locally in each N-LOC blockFor each vignette (12×12 pixels), the *x* coordinate of the keypoint is sent to the N-LOC block.For each pixel, the value of the most active neuron in SL (and its *x* coordinate, i.e., its “line number”) is sent to the relevant N-LOC block.Once all 16 vignettes have been processed, the value of the most active neuron in the PC layer of each N-LOC is sent back to the controller (PS).

As a result, each N-LOC block must:Receive a new compass value every 16 vignettes. The word size for the azimuth buffer takes 8 bits for each period of 16×144 cycles of ref_clk cycle.Receive a new *x* coordinate value for each new vignette. The word size of the Azimuth landmark’s *x* coordinate also takes 8 bits for each 144 period of ref_clk cycle (144 × ref_clk period).Receive a new signature layer value every time a pixel is sent. The word size of signature layer pixels (block’s input) takes 8 bits for each ref_clk cycle.Send its most active place cell value every time a full image has been processed. The word size of the place cell (block’s output) takes 8 bits for each (16×144+cst) ref_clk cycle. cst is a constant which varies with each target system.

#### 6.3.2. Extending N-LOC’s Neural Network on Demand: Leveraging Dynamic Partial Reconfiguration

In order to grant our localisation system better capabilities in term of accuracy-based wide-range navigation, we need to expand the bio-inspired neural network architecture implemented as hardware-accelerator-based N-LOC. However, the resource consumption, and memory footprint of that purpose, is very costly. As we have seen in Table 4, the percentage of resources is limited over ≈180 neuron place cell for each FPGA tile. Therefore, leveraging dynamic partial reconfiguration (DPR) is essential for implementing scalable neural networks. As implemented, the localisation task is large enough that it will not completely fit into the available reconfigurable fabric. Moreover, the computational needs may change according to the vehicle’s environment, e.g., transitioning from a dense urban area to a rural one, with a possible shift in available light. Thus, the neurons used to decide will not be the same and will yield different weights. As a result, relying on a full hardware solution is neither reasonable nor realistic.

Instead, the system should rely on a light software layer which will provide a scheduling and resource management environment to decide which hardware task and where to allocate within an FPGA. Hence, a major step must be achieved, by providing a layer to provide an API to load and replace hardware tasks.

Future Work. A post-scheduling algorithm’s capabilities must be tested to figure out the best task allocation strategy to achieve real-time navigation using a bio-inspired approach. Thus, using a CPU scheduling system, FPGA accelerators can be managed much more efficiently with more complex strategies, which inevitably optimises and outperforms the acceleration.

## 7. Related Work

Overall, a growing autonomous vehicle market needs to implement tasks such as visual navigation, object detection, etc. Hence, a broad summary with various software and hardware-based implementations, running on CPU, GPU and/or FPGAs, is detailed in various surveys [43,44,45,46,47].

One of the major venues to deal with autonomous vehicle navigation is the use of Machine Learning and Deep Learning. Deploying a Deep Learning model directly to edge devices comes with many advantages compared to traditional cloud deployments: by eliminating communications, inter- and intra-processes can reduce latency and reliance on the network connection. Since the data never leave the device, edge-inference helps with maintaining user privacy. Moreover, since the amount of cloud resources is drastically reduced, edge-inference can also reduce ongoing costs [48,49].

The porting of ML applications running on edge devices both drives and is driven by the development of specialised hardware accelerators such as GPUs, ASICs or FPGAs. FPGAs are dominating and attracting people to this research domain, thanks to their steadily improving performance, internal expanded bandwidth and high throughput [50]. A lot of works and benchmark instances are proposed to implement CNN, NN circuits with all required features for, e.g., Xilinx FPGA platforms [51,52], which define a benchmarking approach to co-design, construct and optimise any such algorithm into an inference accelerator IP [53,54].

Another way to implement the navigation process is to resort to bio-inspired models and algorithms. In this context, spiking neural networks (SNNs) and visual place recognition (VPR) models serve different purposes. SNNs, such as the temporal neural encoder (TNE) proposed by Kheradpisheh et al. [55], are inspired by the spiking neurons in the brain and can encode sensory information in the form of spike trains. This allows SNNs to process and recognise temporal patterns and sequences, which are particularly useful for navigation tasks that require tracking of moving objects or path integration. LPMP, as currently designed, does not include temporal sequences (yet), but provides a much simpler model, which in turn makes it easier to follow a hardware–software co-design approach, like the one we used for this work, as the complexity of the neural network is lessened compared to SNNs. VPR models are particularly useful for global localisation tasks in which the robot needs to determine its position relative to a known map of the environment [56]. Hence, our approach relies on a bio-inspired VPR model, which, by contrast with ML/DL models, has a “neural circuitry” which is closer to what can be found in nature, i.e., the way we model individual neurons is not significantly closer to what DL models do, but the structure of the network itself follows more closely what can be found in a mammal’s brain: there are no hidden layers, etc. The resulting neural network is simpler in its structure, but may result in a less memory-efficient way of storing information if implemented naïvely. Hence, our approach relies on a bio-inspired VPR model, which, by contrast with ML/DL models, has a “neural circuitry” which is closer to what can be found in nature, i.e., the way we model individual neurons is not significantly closer to what DL models do, but the structure of the network itself follows more closely what can be found in a mammal’s brain: there are no hidden layers, etc. The resulting neural network is simpler in its structure, but may result in a less memory-efficient way of storing information if implemented naïvely. The LPMP approach (and its hardware implementation) also differs from more “traditional” bio-inspired spiking algorithms in that it relies on recognising visual similarities.

Cuperlier et al. have shown how it could be beneficial to implement a neural processing unit as an IP onto FPGA-based reconfigurable fabrics for an embedded navigation application [57,58]. This is what led us to propose a hardware-based implementation of their bio-inspired algorithm.

Beyond the use of an accurate and precise model, there is the question of providing an implementation that is sufficiently fast to be useful in real life. Hence, the use of accelerators such as GPUs and FPGAs is an important area of research for navigation algorithm implementation to be embedded in vehicles, with performance and energy-efficiency in mind. Qasaimehet et al. in [36] conducted a comprehensive benchmark of the run-time performance and energy efficiency of a wide range of vision kernels in order to determine which embedded platform is most suitable for their application. The conducted study is performed for three commonly used hardware accelerators for embedded vision applications, ARM57 CPU, Jetson TX2 GPU and ZCU102 FPGA using the vendor-optimised vision libraries OpenCV, VisionWorks and xfOpenCV. The results show that the GPU achieves an energy/frame reduction ratio of 1.1×–3.2× compared to the others for simple kernels. However, for more complex kernels and more complete vision pipelines, the FPGA outperforms the others with energy/frame reduction ratios of 1.2×–22.3×. They report also that the FPGA performs increasingly better as a vision application’s pipeline complexity grows.

A publicly available chart summarising neural network accelerator performance and power consumption has been made available by the Energy Efficient Computing Group at Tsinghua University, China [59]. It would be interesting to see where our system fits in this chart.

## 8. Conclusions

We proposed a low-footprint and high-performance accelerator for feature and image recognition in the context of autonomous vehicle navigation. It implements the Spatial Working Memory aspect of the navigation process, along with winner-takes-all implementation to select the most important feature within a pixel stream collected from another IP which was ported for the target FPGA platform and relies on difference of Gaussian operations and Log-Polar representations. Compared to previous (highly accurate) implementations, ours provides not only accuracy, but also very low latency (9× shorter than the baseline reference implementation), low power consumption (up to 5.5× lower) and high frame rate (7× higher). In addition, our experimental results show that the proposed accelerator yields a much lower power consumption footprint (0.257 W for the LPMP implementation; 2.741 W for the whole system) compared to the pure software reference implementation running on a high-end embedded system. We demonstrated Wizarde’s multi-FPGA capability to implement the whole initial neural network (≈10,000 neurons) over multiple tiles, by leveraging its gigabit transceivers. The software processing part will be deployed on the FPGA center tile to communicate and control all FPGA tiles, by receiving and assessing the localisation score of each captured image from different NLOC accelerator modules.

Future work includes implementing LPMP on the Jetson TX2’s GPU, as well as modifying our architecture to increase its clock frequency to improve its performance and compare it to GPU-based embedded systems in terms of performance and power consumption. We also aim to embed our bio-inspired neural IP into a mobile robot to test its limits and perform a runtime assessment of the implemented navigation approach. Furthermore, a dynamic scheduling scenario based on a pre-existing software platform will be proposed to efficiently deploy the whole application by delivering a high performance run-time circuit.

## Figures and Tables

**Figure 1 sensors-23-04631-f001:**
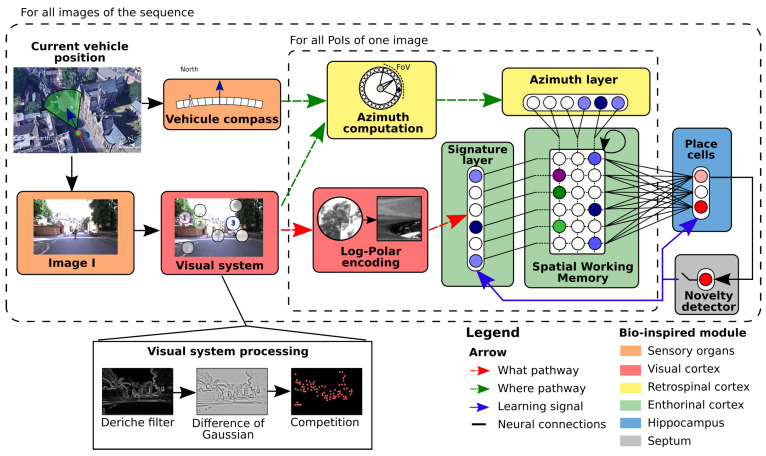
Overview of LPMP model. This figure illustrates how the LPMP model builds a neural representation of an environment from visual information. To do so, the system must go through several stages: points of interest detection (“Visual System”, on the left); saliency points filtering (Deriche filter and DoG); points of interest encoding (Log-Polar encoding); and finally, memory querying (access through signature layer, Spatial Working Memory and place cells). At this stage, the neural activity of Place memory provides the best recognised location based on what it has previously learned.

**Figure 2 sensors-23-04631-f002:**
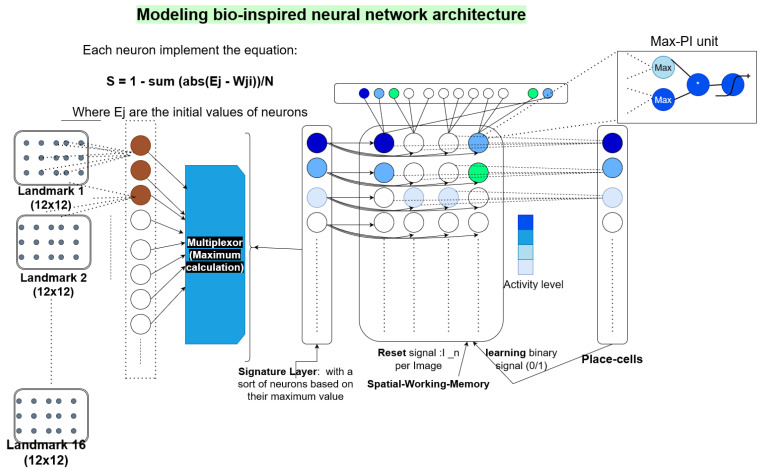
Overview of our bio-inspired neural architecture. On the left-hand side, the image-processing IP identifies and sends keypoint information to the bio-inspired neural IP, pictured on the right-hand side of the figure. Data are sent pixel by pixel, for each landmark of each image. The Learning Mode processes 10 images for each time period. If a consensus is found (i.e., the measured error is acceptable), the system then switches to using mode.

**Figure 3 sensors-23-04631-f003:**
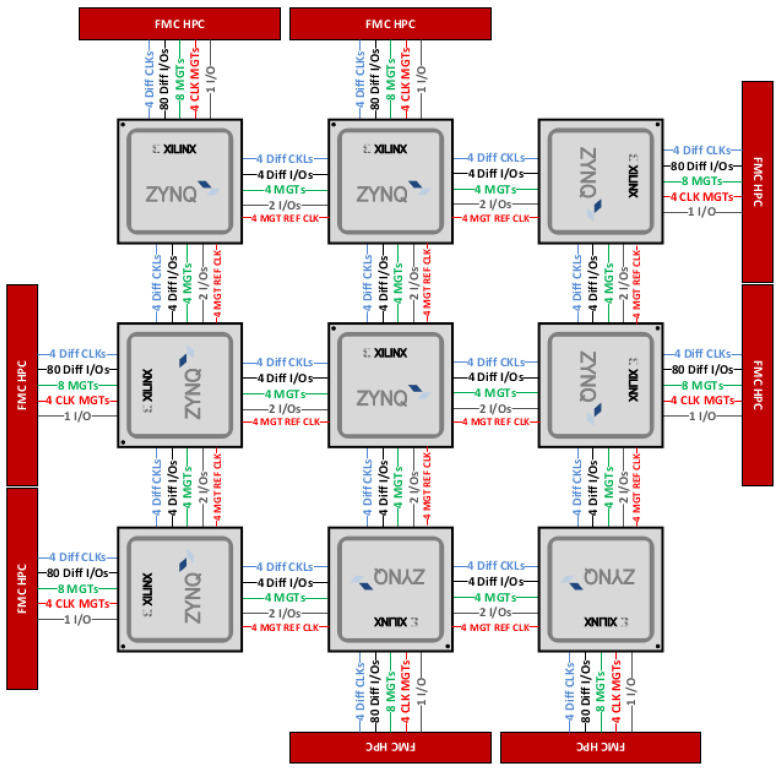
The Wizarde Platform. Each tile is connected to the others through gigabit transceivers (see Table 2).

**Figure 5 sensors-23-04631-f005:**
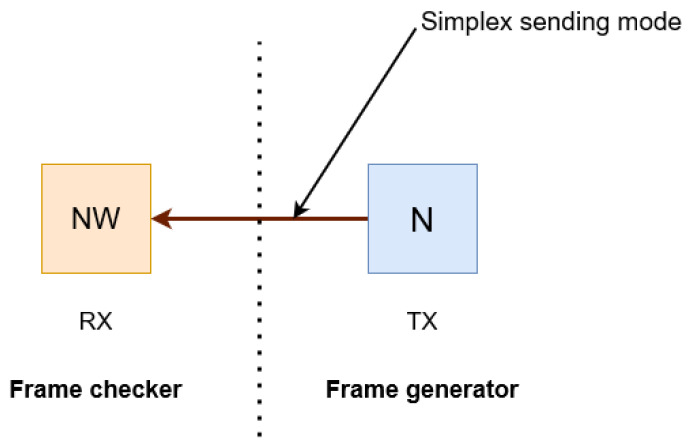
A simplex mode implemented on GTX transceivers as first-stage protocol to communicate between two adjacent Wizarde tiles.

**Figure 6 sensors-23-04631-f006:**
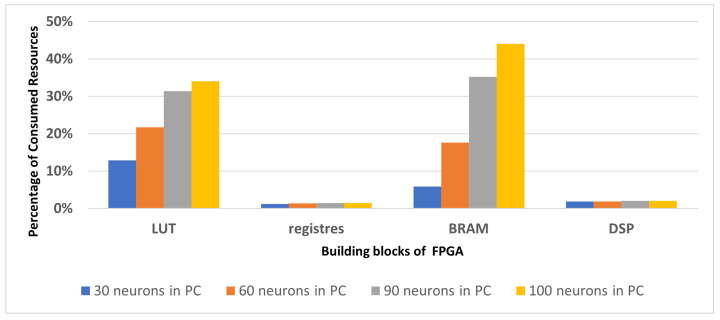
Resource consumption in term of LUT, BRAM, registers and DSP, for configurations with different numbers of place cells.

**Figure 7 sensors-23-04631-f007:**
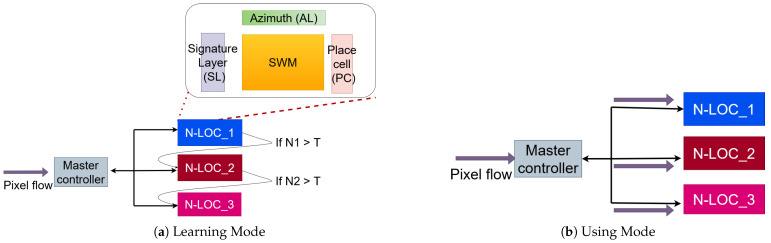
Expanding bio-inspired neural network architecture N-LOC over Wizarde multi-tiles. In the learning phase, the data-copy of each current landmark will be held on one bloc IP. If the number of learned neurons in N-LOC1 is overloaded (superior to a fixed threshold *T*), we will switch into next available bloc IP which is the second one. Then, it will be the same rule for all different bloc IPs. In process phase, all bloc IPs works simultaneously, the best score among the three represents the accurate and appropriate localisation of a given image.

**Figure 8 sensors-23-04631-f008:**
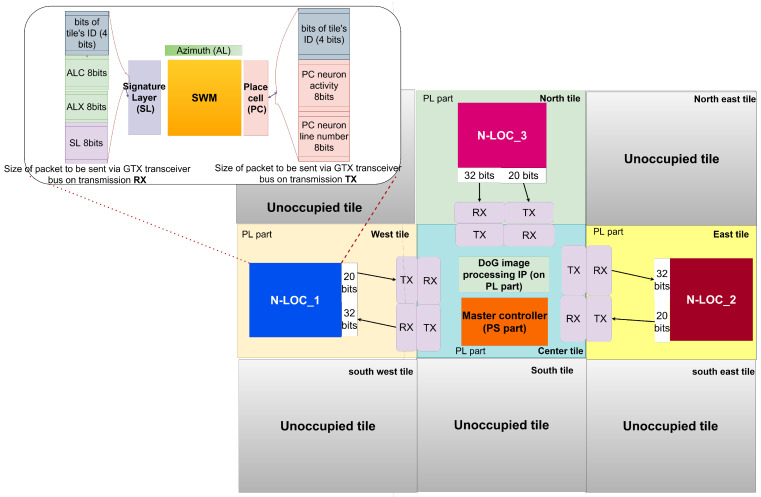
Overall architecture for a design leveraging 3 N-LOC instances. The design is implemented throughout multi-N-LOC architecture based on Wizarde (see Section 2). TX/RX pairs can be implemented following multiple means: gigabit transceivers, GPIOs, Ethernet, etc. An N-LOC IP (detailed in the upper-left corner) awaits an *x* Azimuth coordinate (i.e., its row number), an *x* landmark coordinate (i.e., also its row number), the current pixel to process and its tile ID within Wizarde. Conversely, an N-LOC instance sends the score obtained in the local WTA, its line number in the local Place Cell Memory and its tile ID. The image acquisition and processing IP is implemented in the central tile and a lightweight resource manager collects local WTA winners and performs the final WTA, in parallel with scheduling communications.

**Figure 9 sensors-23-04631-f009:**
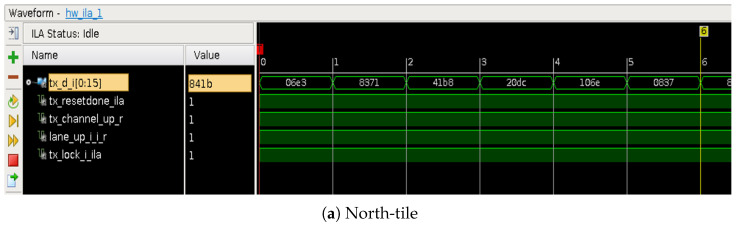
Waveform data acquisition. We trigger the data acquisition on the chipscope from the pseudo-random value 0x06E3 and we can verify that the reset and activation signals of the GTX are valid and the received data are in conformity with those sent. We also check the error-accumulator signal remains at 0. This benchmark is set up with a throughput of 6.25 Gbps.

**Figure 10 sensors-23-04631-f010:**
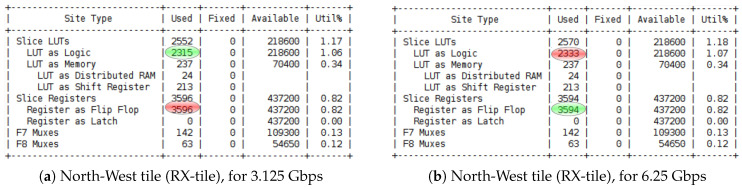
Vivado report: Resource usage per tile. The usage rates are almost equal; as such, 18 more logic LUTs and 2 less Flip Flops were recruited for the 6.25 Gbps frequency upgrade.

**Figure 11 sensors-23-04631-f011:**
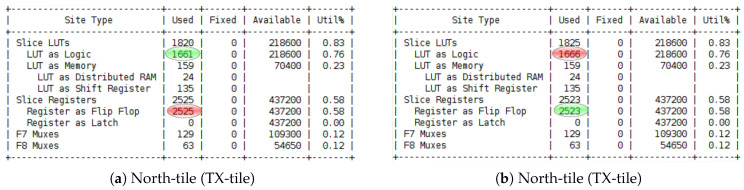
Vivado report: Resource usage per tile. The usage rates are almost equal; as such, 5 more logic LUTs and 2 less Flip Flops were recruited for the 6.25 Gbps frequency upgrade.

**Table 1 sensors-23-04631-t001:** FPGA vs. GPU comparison: Host-Accelerator data transfer overhead, in milliseconds. The latency was computed using 1000 roundtrips, each exchanging a 32-bit value.

	Zynq-7045	Nvidia Jetson TX2	Nvidia Jetson Xavier AGX
Overhead (ms)	$7.02\times 10^{-3}$	$5.70\times 10^{-1}$	$1.28\times 10^{-1}$

**Table 2 sensors-23-04631-t002:** Contents of a Wizarde tile, based on the Zynq xc7z045ffg900-2.

Zynq SoC	A9 + Kintex-7
LUTs	218,600
FlipFlops	437,200
Block RAMs	545
SMA connector	1 port
DDR3 SDRAM	1 GB (connected to PS)
DDR3 SODIMM	1 GB (connected to PL)
Gbit Transceivers	16
USB 2.0/UART	1 port
Gbit Ethernet	1 port

**Table 3 sensors-23-04631-t003:** Resource usage as a function of the number of landmarks used per image processed in Fiack et al.’s design. Results obtained on a port carried out on Wizarde with Vivado 2016.4.

Max. # Landmarks/Img	8	16	32	48
LUTs (%)	8.89	13.56	25.3	39.91
Registers (%)	4.46	6.8	13.68	23.58
BRAMs (%)	38.07	46.88	64.5	82.11
DSPs (%)	21.56	32.22	53.56	74.89

**Table 4 sensors-23-04631-t004:** Resource consumption estimation for 8, 16 and 32 landmarks, depending on the number of neurons in the place cell layer. Only 50 and 80 neurons are shown for 32 landmarks: resources saturate beyond that number.

# Neurons	50	80	90	100	150	170	180
**8 Landmarks**							
LUTs (%)	11.68	15.56	18.08	19.79	29.12	31.99	27.41
BRAMs (%)	16.70	31.38	49.72	53.39	86.42	93.76	132.29
FLIP FLOP (%)	2.69	1.67	2.85	2.89	3.16	3.27	2.03
DSPs (%)	1.33	0.33	1.33	1.33	1.33	1.33	1.44
**16 landmarks**							
LUTs (%)	16.73	24.51	28.37	31.41	41.75	46.67	52.32
BRAMs (%)	35.05	60.73	97.43	104.77	110.28	124.95	150.41
FLIP FLOP (%)	3.21	2.55	3.45	3.50	2.13	4.16	4.76
DSPs (%)	1.33	0.33	1.33	1.33	1.67	2.37	2.89
**32 landmarks**							
LUTs (%)	33.62	48.21	55.45	60.90	88.19	99.08	104.53
BRAMs (%)	24.04	94.50	94.50	94.50	376.33	376.33	376.33
FLIP FLOP (%)	2.68	1.33	2.71	2.71	2.75	2.75	2.76
DSPs (%)	1.89	2.44	1.89	1.89	1.89	1.89	1.89

**Table 5 sensors-23-04631-t005:** N-LOC: Performance and efficiency. The system is configured with different numbers of place cell neurons, corresponding to the number of neurons to be learned. The system is then evaluated and tested with 100 images for different place cell configurations. Results are generated using Nvidia Jetson TX2 platform for software reference (using all 6 CPU cores when possible for the optimised version) and an FPGA ZC706 board for N-LOC Hardware implementation. The used frequency in FPGA in both 30 and 60 PC neurons is 100 MHZ. For 90 neurons, the frequency is set it 70 MHZ to satisfy timing constraints.

Number of Neurons in Place Cells	30	60	90
*N-LOC*			
*Learning*: Latency (ms)	2.6	2.6	2.6
*Using*: Latency (ms)	11.4	20.3	29.2
Total Throughput (img/s)	82	45	31
Total Power consumption	≈2.8	≈2.8	≈2.8
static + dynamic (W)			
*Baseline reference*			
*Learning*: Latency (ms)	17.76	17.86	18.99
*Using*: Latency (ms)	100.32	123.93	146.66
Total Throughput (img/s)	9	7	6
*Optimised reference*			
*Learning*: Latency (ms)	17.16	17.10	17.48
*Using*: Latency (ms)	61.11	66.05	72.60
Total Throughput (img/s)	13	12	11
Total Power consumption	≈7.5–15	≈7.5–15	≈7.5–15
static + dynamic (W)			

**Table 6 sensors-23-04631-t006:** Resource utilisation for each integrated IP, implemented on one tile of Wizarde, for 100 neurons of place cell neuron group and 16 landmarks per each image. The raw values are provided, along with the percentage they represent between parentheses.

IP Block	Slice LUTs	BRAM Tile	FLIP FLOPs	DSPs
Image processing	29,647 (13.6%)	255 (46.8%)	30,553 (7.0%)	298 (33.1%)
Single N-LOC	62,377 (28.5%)	164 (30.0%)	7879 (1.8%)	4 (0.4%)

**Table 7 sensors-23-04631-t007:** Total power consumption (static + dynamic) generated by all integrated IPs. Results obtained with 100 place cell neurons (maximum of neurons to be trained) and 16 landmarks per image.

(IP) Block	Description	Power (W)
Image processing	Image acquisition and landmark identification	0.783
Bio-inspired neural accelerator	Neuron activation; place cell recognition	0.141
Processing system	Hardcore processor (ARM Cortex A9)	1.567
Total power on chip (static + dynamic)		2.749

**Table 8 sensors-23-04631-t008:** Timings for 90 place cell neurons: Python reference code (“Baseline Ref”), optimised multicore version (“Optimised Ref”), a single large N-LOC instance, and a distributed 3 × N-LOC architecture (3×30 neurons), implemented on a single Zynq-7045 SoC’s FPGA part. The controller is implemented on the Cortex A9 as bare-metal software.

	Baseline Ref	Optimised Ref	1×90 N-LOC	3×30 N-LOC
Learning Latency (ms)	18.99	17.6	2.6	3.57
Using Latency (ms)	146.66	72.60	29.2	9.81
Total Throughput (img/s)	6	11	31	70

**Table 9 sensors-23-04631-t009:** Single and distributed N-LOC: speedups. Baseline: the optimised reference Python application compiled with Cython.

Speedups	1×90 N-LOC	3×30 N-LOC
Learning Latency	8	4
Using Latency	3	7
Total Throughput	4	6

**Table 10 sensors-23-04631-t010:** N-LOC: Power consumption (static + dynamic) (in Watts) for 90 place cell neurons. The last column computes the power consumption ratio between a 1×90 and a 3×30 configuration.

IP Block	1×90 N-LOC	3×30 N-LOC	3×30 vs. 1×90
N-LOC	0.472	0.993	2.10
Processing System	1.629	1.639	1.00
TOTAL	2.101	2.638	1.25

**Table 11 sensors-23-04631-t011:** N-LOC: resource usage. The percentage of a given resource usage on the Zynq-7045 is given between parentheses. Each processed image contains 16 landmarks.

# Neurons	Slice LUTs	BRAM Tile	FLIP FLOPs	DSPs
1×90 N-LOC	68,209 (31.2%)	192 (35.2%)	3039 (0.7%)	18 (2.0%)
1×30 N-LOC	28,869 (13.2%)	32 (5.9%)	2614 (0.6%)	17 (3.4%)
3×30 N-LOC	84,198 (38.51%)	96 (17.61%)	8199 (3.75%)	51 (10.2%)

**Table 12 sensors-23-04631-t012:** N-LOC: Performance (number of operations per second) per Watt.

1×100 Ref	769 KOPS
1×100 N-LOC	239 MOPs
1×90 Ref	820 KOPS
1×90 N-LOC	269 MOPs
3×30 N-LOC	299 MOPs

**Table 13 sensors-23-04631-t013:** Aurora IP configuration. The line transmission rate is set to 6.25 Gbps and the GT reference clock is set to 125 Mhz on both TX and RX of adjacent tiles (north and north-west).

	North FPGA	North-W FPGA
Lane Width (Bytes)	2	2
Line Rate (Gbps)	6.25	6.25
GT Refclk (Mhz)	125	125
Init clk (Mhz)	50	50
DRP clk (Mhz)	50	50
DRP clk (Mhz)	TX-only	RX-only
	simplex	simplex

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
