# Peer review of "Implementation of a Bio-Inspired Neural Architecture for Autonomous Vehicles on a Multi-FPGA Platform†"

_sensors, 2023, doi:10.3390/s23104631_

Round 1

Reviewer 1 Report

This paper presents a bio-inspired neural architecture implementation on FPGA. Such a system can be used for localization on autonomous vehicles, with the advantages of using inexpensive and reliable camera imaging sensors instead of expensive LiDAR sensors or satellite positioning services. The authors presented the background information with abundant details and readers who are not familiar with the context can follow without problems. The overall approach is sound, and the presentation is clear. The achieved latency reduction and throughput improvement are promising.

One thing that the authors might want to improve is the discussion on the accuracy of their FPGA implementation. The authors mentioned that they have to reduce the number of neurons and use fixed-point numbers on FPGAs. How these affect the accuracy of the overall system is not sufficiently discussed. Whether the implementation is evaluated on-board or based on simulation/emulation is also not clearly stated. Besides, the Vivado 2016.4 is very old (7 years!) and some justification/explanation would be helpful.

There are some other minor typo/grammar issues:

Line 131, "a fast lanes"

Line 234, "Thus, To"

Line 397, "biimplementationo-inspired"

Line 483, "numer"

Author Response

Original Manuscript ID: sensors-2342761

Original Article Title: “Implementation of a bio-inspired neural architecture for autonomous vehicle on a multi-FPGAs platform

To: MDPI Sensors (ISSN 1424-8220)

Re: Response to reviewers

Dear Editor,

Thank you for allowing a resubmission of our manuscript, with an opportunity to address the reviewers’ comments.

We are uploading (a) our point-by-point response to the comments (below) (response to reviewers), (b) an updated manuscript with “track-changes” function indicating changes (Supplementary Material for Review).

Comments from Reviewer 1:

1- There are some other minor typo/grammar issues.

We thank the reviewers for their comments. We have fixed the outlined typos.

Comment: One thing that the authors might want to improve is the discussion on the accuracy of their FPGA implementation. The authors mentioned that they have to reduce the number of neurons and use fixed-point numbers on FPGAs. How these affect the accuracy of the overall system is not sufficiently discussed.

Answer: We did experiment with varying resolutions for fixed-point numbers. However we saw no accuracy improvement in our experiments. We have added a sentence to mention this in Section 5.3

Comment: Whether the implementation is evaluated on-board or based on simulation/emulation is also not clearly stated. Besides, the Vivado 2016.4 is very old (7 years!) and some justification/explanation would be helpful.

Answer: The Wizarde platform was validated using Vivado 2016.4 by the contractor which actually fabricated the board according to our design, and we haven’t updated the Linux image with the appropriate device tree for a newer version. However, we plan to do so in the near future. While a newer version of Vivado may better utilise resources, the number of available DSPs (used by the winner-take-all part of the architecture) remains constant on a single tile, so the resulting resource occupation will likely not change by much. A quick synthesis using Vivado 2019.3 yields a difference lower than 4% in terms of resource occupation. We have added text to the Experimental Setup section to clarify this.

Best regards,

Tarek Elouaret et al.

Reviewer 2 Report

For an autonomous vehicle camera, the authors proposed a method to implement an energy-efficient visual localization model on an FPGA platform. Overall, the authors have made a good attempt. This is well written and organized paper. It is scientifically sound and contains sufficient interest to merit publication. However, due to the lack of comparison data with specific data, the effectiveness of the proposed technique is not clear. My other comments are as follows:

1.      The authors should not use acronym without explanation. All acronyms must be defined before use. For example, FPGA (Field Programmable gate Array), and so on.

2.      The research survey is not enough. The articles discussed in the introduction part are out of date. (Only Ref. [23] is a new article.) For example, the papers published within 2 years are used to calculate CiteScore in SCOPUS. The authors should survey past studies in detail.

3.      In section 5.5, the experimental setup is not clear. The authors should demonstrate the experimental setup with some pictures. This seems like a simulation, not an experiment.

4.      Table 4 is out of frame. The authors should improve the presentation.

5.      There is no Y-axis label in Fig. 6 and 8.

6.      In section 7, the authors' proposed method does not adequately describe their data. This interpretation is not supported by any demonstrations. Readers will fail to understand the scientific contribution of this research. The authors should justify the authors’ opinion with specific data.

Author Response

Original Manuscript ID: sensors-2342761

Original Article Title: “Implementation of a bio-inspired neural architecture for autonomous vehicle on a multi-FPGAs platform

To: MDPI Sensors (ISSN 1424-8220)

Re: Response to reviewers

Dear Editor,

Thank you for allowing a resubmission of our manuscript, with an opportunity to address the reviewers’ comments.

We are uploading (a) our point-by-point response to the comments (below) (response to reviewers), (b) an updated manuscript with “track-changes” function indicating changes (Supplementary Material for Review).

Comments from Reviewer 2:

Comment: However, due to the lack of comparison data with specific data, the effectiveness of the proposed technique is not clear.

1- The authors should not use acronym without explanation. All acronyms must be defined before use. For example, FPGA (Field Programmable gate Array), and so on.

We thank the reviewer for pointing this out. We made sure that all acronyms are listed under the Abbreviations title.

2-  The research survey is not enough. The articles discussed in the introduction part are out of date. (Only Ref. [23] is a new article.) For example, the papers published within 2 years are used to calculate CiteScore in SCOPUS. The authors should survey past studies in detail.

We thank the reviewer for pointing this out. . We have added several more recent citations. However, we would also like to point out that many cited references (roughly between a third and a quarter of all the references) were published in 2020 and 2021, i.e., less than 4 years ago, and are still very relevant to the work we submitted.

3-   In section 5.5, the experimental setup is not clear. The authors should demonstrate the experimental setup with some pictures. This seems like a simulation, not an experiment.

We  thank again the reviewer for bringing up this comment.  We agree the experimental setup was not clear enough. We have added text to clarify our explanations pertaining to the experimental testbed, setup, and protocol. In particular, we made sure to make it clear we did implement/synthesise the IP on an actual FPGA (we did not simulate anything). However, it is unclear what kind of pictures would help in that case: as we are using the Oxford dataset to feed N-LOC, the only output would be on a computer screen. Likewise, we could add a picture of the Wizarde board, but we fail to see how that would help with the experiments description.

4-  Table 4 is out of frame. The authors should improve the presentation.

We thank again the reviewer for noticing this oversight. We fixed the mistake in the paper.

5-  There is no Y-axis label in Fig. 6 and 8.

We thank the reviewer for this comment. We reviewed all of our figures to make sure axes were properly labeld, including Figure 6.

6-  In section 7, the authors' proposed method does not adequately describe their data. This interpretation is not supported by any demonstrations. Readers will fail to understand the scientific contribution of this research. The authors should justify the authors’ opinion with specific data.

We agree Section 7 needed some reorganization and some rephrasing to clarify our intent. Section 7 is meant to provide a broad view of other methodologies which attempt to tackle the same issues we are trying to solve: autonomous vehicles navigation, and how to accelerate it. Our point is to show how ML/DNN rely on fundamentally different assumptions, and not that our current implementation outperforms those DNN/ML implementations that currently exist. Likewise, there are several approaches to bio-inspired navigation. We have added some text to this section to describe how the original LPMP approach differs from Spike algorithms, which are also bio-inspired. Further, as we describe in our introduction, our goal is to accelerate the LPMP model to make it viable in a real-time context for high speeds. The validity of the bio-inspired LPMP algorithm (in terms of accuracy and precision) has already been demonstrated by Espada et al., and later confirmed with improvements by Colomer et al., 2021, Colomer et al. and with additional real-life environment results in S.Colomer’s Ph.D. dissertation (defended early 2023). The latter features  features results obtained in a real-life environment (closed tracks for a mobile robot to run LPMP). Hence, while the work shown in this paper “only” resorts to datasets, the reference code was run in a real-life situation, and correctly identified landmarks. Hence, since both implementations of the model select the same landmarks in our dataset tests, and since the reference implementation also detects the correct landmarks in a real-life context, we consider that our FPGA-based implementation is faithful to the LPMP model. Additional text has been inserted in the experimental setup section to clarify this point.

Hence, our only goal is to explore how to properly and efficiently implement LPMP in hardware so that it performs well with a low power footprint. This is why we only compare ourselves to the baseline software reference.

Regarding a GPU implementation, it is on our TO-DO list to implement a GPU-based solution which ports LPMP on the Jetson TX2 platform we already used for reference code purposes. We also plan to install our system in a mobile robot to perform the same types of measurements that were done with the reference code. However, it should be noted this is going far beyond the scope of the current paper, and installing such systems in mobile robots require additional configuration, as well as additional programming , and in general tuning, before tests can be run.

The newly added references are as follows: 9, 40, 42, 45, 48, 56, 57

Best regards,

Tarek Elouaret et al.

Reviewer 3 Report

Autonomous vehicles require professional self-localization mechanisms, and cameras are widely used sensors due to their cost-effectiveness and wide input capabilities. However, the computational load of visual localization is environment dependent and requires real-time processing and power efficient decision making. Field Programmable Gate Arrays (FPGAs) provide a solution for prototyping and energy efficiency evaluation. In this study, the authors propose a distributed solution to implement a large bio-inspired visual localization model. The proposed workflow includes an intellectual property (IP) module for image processing that provides pixel information for each visual cue found in each captured image; implementation of N-LOC, a biology-based neural architecture, on an FPGA board and a distributed version of N-LOC with evaluation on a single FPGA and design for use on a multi-FPGA platform. The proposed hardware implementation of IP demonstrates up to 9 times lower latency and 7 times higher throughput (frames per second) while maintaining energy efficiency compared to a pure software solution. The proposed system consumes 2.741 watts for the entire system, which is 5.5-6 times less than the average consumption of the Nvidia Jetson TX2.

Despite the satisfactory quality of the article, there are some shortcomings that need to be corrected.

  1. The aim of the paper should be defined.
  2. The novelty of the proposed approach should be clearly defined.
  3. The potential limitations of the proposed solution, such as the types of environments or lighting conditions in which the visual localization model might not perform as effectively, should be discussed.
  4. Formulas are parts of the sentences. The punctuation should be corrected.

In summarizing my comments I recommend that the manuscript is accepted after minor revision. 

Author Response

Original Manuscript ID: sensors-2342761

Original Article Title: “Implementation of a bio-inspired neural architecture for autonomous vehicle on a multi-FPGAs platform

To: MDPI Sensors (ISSN 1424-8220)

Re: Response to reviewers

Dear Editor,

Thank you for allowing a resubmission of our manuscript, with an opportunity to address the reviewers’ comments.

We are uploading (a) our point-by-point response to the comments (below) (response to reviewers), (b) an updated manuscript with “track-changes” function indicating changes (Supplementary Material for Review).

Comments from Reviewer 3:

1- The aim of the paper should be defined.

We thank the reviewer for bringing this to attention. We have added text in the fifth and sixth paragraphs of the introduction to make the aim of the paper more explicit.

2-  The novelty of the proposed approach should be clearly defined.

We thank the reviewer for pointing this out. We have made an explicit list of contributions in this paper, including differences compared to the previous publication. 

3-   The potential limitations of the proposed solution, such as the types of environments or lighting conditions in which the visual localization model might not perform as effectively, should be discussed.

We  thank again the reviewer for bringing up this comment. The solution we propose is an implementation of the LPMP model on FPGA (software to hardware). In our study, we tested the model based on the LPMP (frontiers) paper by Colomer et al. (2021), using different lighting and navigation conditions. We have added a subsection titled “Advantages and Limitations of LPMP” in the Background section to outline where the model works well, and where it has shortcomings so far.

However, we plan to conduct additional tests to improve the model's performance, but this is out of scope for this paper.Here is the link of Colomer’s article (https://www.frontiersin.org/articles/10.3389/frobt.2021.703811/full).

4-     Formulas are parts of the sentences. The punctuation should be corrected.

Thank you once again for your comment. To the best of our knowledge, we have fixed any punctuation issue with our equations.

Best regards,

Tarek Elouaret et al.
